# Bridging the Gap: A Systematic Review with Expert Opinion on the Use of Dalbavancin for In-Label and Off-Label Indications in Pediatric Patients

**DOI:** 10.3390/antibiotics14020121

**Published:** 2025-01-23

**Authors:** Désirée Caselli, Maurizio Aricò, Elio Castagnola, Milo Gatti

**Affiliations:** 1Department of Consorziale Policlinico di Bari, Ospedale Pediatrico Giovanni XXIII, 70100 Bari, Italy; desiree.caselli@policlinico.ba.it; 2Department of Pediatria, Azienda Sanitaria Locale di Pescara, 65020 Pescara, Italy; maurizio.arico@asl.pe.it; 3Infectious Disease Unit, IRCCS Istituto Giannina Gaslini, 16159 Genova, Italy; eliocastagnola@gaslini.org; 4Department of Medical and Surgical Sciences, Alma Mater Studiorum University of Bologna, 40100 Bologna, Italy; 5Department of Clinical Pharmacology Unit, Integrated Infectious Risk Management, IRCCS Azienda Ospedaliero-Universitaria di Bologna, 40100 Bologna, Italy

**Keywords:** dalbavancin, pediatric patients, acute bacterial skin and skin structure infections, bone and joint infections, central line-associated bloodstream infections, pharmacokinetic/pharmacodynamic

## Abstract

Objectives: The aim of this work was to perform a systematic review assessing the pharmacokinetic/pharmacodynamic (PK/PD) properties of dalbavancin and the clinical use for in-label and off-label indications in pediatric patients. Methods: Two authors independently searched the PubMed-MEDLINE and Scopus databases and clinicaltrials.gov up to 20 November 2024, to retrieve randomized controlled trials (RCTs), observational studies, PK studies, and case series/reports assessing dalbavancin PK/PD properties or the clinical use for both in-label and off-label indications in pediatric patients. Data were independently extracted by the two authors, and the quality of the included studies was independently assessed by means of specific tools according to study design. Clinical success was selected as the primary outcome. Descriptive statistics were used for summarizing the retrieved data. Subgroup analysis according to PK/PD data, as well as in-label and off-label indications, was performed. Results: After screening 206 articles, nine studies were included in the systematic review (one RCT, three PK studies, and five case series/reports; *n* = 267). Dalbavancin exposure was 30% lower in pediatric patients compared to adults. In acute bacterial skin and skin structure infections (ABSSSIs), the overall clinical success of dalbavancin was 96.1-97.3% and 92.9% in RCT and case series, respectively. Bone and joint infections (60.7%) and central-line-associated bloodstream infections (14.3%) represented the most common dalbavancin off-label indications in pediatric patients. Overall, the clinical success for off-label indications was 92.9%. The rate of adverse events ranged from 7.1% to 10.7%. Conclusions: Our systematic review summarized evidence concerning the PK/PD properties of dalbavancin and its use for in-label or off-label indications in pediatric patients. The available findings suggest that dalbavancin may be a valuable alternative for the management of ABSSSIs and/or off-label indications in pediatric patients according to efficacy and safety data, allowing for a potential minimized duration of hospital stay.

## 1. Introduction

Dalbavancin is a novel long-acting lipoglycopeptide characterized by its long elimination half-life, good tissue penetration in different sites (including skin, bone, and epithelial lining fluid), excellent activity against Gram-positive pathogens (including resistant strains such as methicillin-resistant *Staphylococcus aureus* [MRSA] with MIC_50_ and MIC_90_ values of 0.03 and 0.06 mg/L, respectively), and its good anti-biofilm activity [1,2,3]. Dalbavancin exhibits concentration-dependent activity, and the ratio between the free area under the time-to-concentration curve and the minimum inhibitory concentration (*f*AUC_24_/MIC) was established as the best pharmacodynamic (PD) predictor of clinical success, granting an *f*AUC_24_/MIC > 111.1 a 2-log kill activity against *Staphylococcus aureus* [1]. Thanks to these properties, dalbavancin possesses long-term efficacy despite the simplified weekly administration regimens, allowing for a reduction in unnecessary daily in-hospital intravenous antibiotic regimens in the management of suspected and/or documented Gram-positive infections [4,5]. 

Dalbavancin was firstly approved by the Food and Drug Administration (FDA) and by the European Medicines Agency (EMA) in 2014 and 2015, respectively, for the management of acute bacterial skin and skin structure infections (ABSSSIs) in adult patients at the dosing regimen of 1500 mg administered as either a single infusion of 1500 mg or as 1000 mg followed one week later by 500 mg. Subsequently, in 2021 and 2022, the FDA and EMA, respectively, extended the indication for dalbavancin use to pediatric patients aged at least 3 months according to the results of phase I studies and preliminary findings retrieved in a phase III pivotal trial [6,7,8].

Although the use of dalbavancin for in- and off-label indications has been well characterized in recent years [4,9,10,11,12], evidence for the use of this agent in pediatric patients remains limited, still representing an unmet clinical need [13]. Furthermore, its well-recognized pharmacokinetic/pharmacodynamic (PK/PD) profile makes dalbavancin a promising alternative for the management of ABSSSIs, bone and joint infections, and/or other off-label indications in pediatric scenarios.

The aim of this study was to perform a systematic review assessing the PK/PD properties of dalbavancin and the clinical use for in-label and off-label indications in pediatric patients.

## 2. Results

### 2.1. Literature Search

A total of 206 potential studies were retrieved, of which 191 were excluded after initial screening of titles and abstracts, as well as searching for duplicates. Overall, 15 full-text articles were considered potentially eligible, and 9 out of these met the final inclusion criteria. Six additional studies were excluded according to the following reasons: inclusion of adult patients only (*n* = 2); lack of available results (*n* = 2); lack of subgroup analysis for pediatric patients (*n* = 1); and assessment of in vitro susceptibility without outcome data (*n* = 1; Figure 1). 

### 2.2. Features of the Included Studies

The nine included studies consisted of one randomized controlled trial (RCT) [8], four case series [14,15,16,17], one case report [18], two phase I studies [6,7], and one population PK model [19]. Five studies were multicentric [6,7,8,14,19]. Three studies were conducted in Italy [14,16,17], two studies each in the USA [6,7] and in Spain [15,18], and the other two worldwide [8,19]. Three studies each evaluated dalbavancin PK/PD properties in pediatric patients [6,7,19] and in-label indications [8,16,17], two studies assessed the clinical efficacy of dalbavancin for off-label indications [15,18], whereas one study included patients receiving dalbavancin for both in- and off-label indications [14].

Overall, a total of 267 pediatric patients receiving at least one dalbavancin dose were enrolled (204 out of 211 patients included in the study by Carrothers et al. [19] were previously enrolled in another three studies [6,7,8]). The age of the included patients ranged from 4 months to 18 years, with a male preponderance (70.7%). Patients received from 1 up to 16 dalbavancin doses during treatment courses. Combination therapy was adopted in five cases (four and one with rifampicin and cotrimoxazole, respectively).

### 2.3. Dalbavancin PK/PD Features in Pediatric Patients

The main findings of the studies evaluating the PK/PD properties of dalbavancin in pediatric patients are summarized in Table 1. Overall, three studies (two phase 1 PK studies and one population PK model) were found [6,7,19].

Bradley et al. [6] assessed the PK behavior of single-dose dalbavancin (1000 mg in patients >60 kg and 15 mg/kg in those < 60 kg) in ten pediatric patients ranging from 12 to 17 years in age. Overall, a reduction of approximately 30% in the area under the time-to-concentration curve (AUC) was found compared to adult patients, whereas no difference in the volume of distribution was reported.

Gonzalez et al. [7] performed a population PK study to evaluate the PK behavior of single-dose dalbavancin in 33 pediatric patients aged 3 months–11 years and to define the best dosing schedule among pediatric patients. In total, 10 mg/kg, 25 mg/kg, and 15 mg/kg were administered in patients of age 3 months–2 years (median weight 9.6 Kg), 2–5 years (median weight 15.7 Kg), and ≥5 years (median weight of 31.4 Kg and 60.4 Kg for those of 6–11 years and 12–17 years, respectively), respectively. According to model simulations, a single dose of 22.5 mg/kg and 18 mg/kg in patients aged 3 months–5 years and 6–17 years, respectively, achieved a similar dalbavancin exposure to that reported in adults receiving a single dose of 1500 mg.

Carrothers et al. [19] developed a population PK model by including 1124 dalbavancin concentrations collected from 211 pediatric patients. Overall, dalbavancin PK was well characterized by a three-compartment model and the approved pediatric dosing regimens granted a probability of target attainment ≥94% for MIC values ≤2 mg/L and ≤0.5 mg/L for the stasis and 2-log kill targets, respectively.

### 2.4. Dalbavancin Use for In-Label Indications

The main findings of the studies assessing the clinical efficacy of dalbavancin use for in-label indications in pediatric patients are reported in Table 2. Overall, four studies (one RCT and three case series) were retrieved [8,14,16,17], and 189 pediatric patients ranging from 0.04 to 18 years in age (60.8% male) were included. All included pediatric patients received dalbavancin for the management of ABSSSIs, of which six cases (3.2%) were defined as complicated. Methicillin-susceptible *Staphylococcus aureus* (MSSA) was the most frequently isolated pathogen (94 cases; 49.7%), followed by MRSA (14 cases of which were due to community-acquired strains; 7.4%), and *Streptococcus pyogenes* (8 cases; 4.2%).

Giorgobiani et al. [8] randomized 191 patients to receive a single dalbavancin dose (*n* = 83), two dalbavancin doses (*n* = 78), or comparator agents (i.e., vancomycin, oxacillin, or flucloxacillin; *n* = 30) for the management of ABSSSIs. Overall, the clinical response at 48–72 h and at the test of the cure was 97.4% and 96.1% in patients treated with a single dalbavancin dose, 98.6% and 97.3% in those receiving two dalbavancin doses, and 89.7% and 100.0% in the comparator group, respectively. The rate of AEs was 7.2%, 10.3%, and 3.3% in patients treated with a single dalbavancin dose, two dalbavancin doses, or comparator agents, respectively.

In the other three case series [14,16,17], the vast majority of patients (92.9%) received a previous antibiotic regimen. A total of 24 out of 28 patients received only one dalbavancin dose, whereas the other four were treated with two dalbavancin doses. Combination therapy with rifampicin was adopted in two cases (7.1%). Dalbavancin TDM was not performed in any cases.

Overall, clinical cure was reported in 26 patients (92.9%), and no cases of recurrence and/or relapse were documented. AEs occurred in two patients (7.1%), consisting of one case of a headache and vomiting and one showing fever associated with rash/urticaria. Dalbavancin withdrawal was needed in both patients.

### 2.5. Dalbavancin Use for Off-Label Indications

The main findings of the studies assessing the clinical efficacy of dalbavancin use for off-label indications in pediatric patients are summarized in Table 3.

Overall, three studies (two case series and one case report) were retrieved [14,15,18] and 28 pediatric patients ranging in age from 0.3 to 18 years (67.9% male) were included. The main off-label indications were bone and joint infections (BJIs) in 17 cases (60.7%), central-line-associated bloodstream infections (CLABSIs) in four cases (14.3%), surgical site infections (SSIs) in three cases (10.7%), and a case each (3.6%) of community-acquired pneumonia (CAP), septic thrombophlebitis, chronic cellulitis, and chronic infection of a patch used for repairing congenital heart disease. MRSA and MSSA (8 cases each; 28.6%) represented the most frequent pathogens.

The vast majority of patients (96.3%) received a previous antibiotic regimen. The total administered dalbavancin doses ranged from 1 to 16 per patient. Combination therapy was implemented in three cases (10.7%). Dalbavancin TDM was not performed in any cases.

Overall, clinical cure was reported in 26 patients (92.9%), with no cases of relapse and/or recurrence. AEs occurred in three patients (10.7%), leading to dalbavancin withdrawal in two cases.

### 2.6. Quality Assessment

In the only RCT included, some concerns were found in four out of five assessed domains (i.e., randomized process, deviation from the intended interventions, measurement of the outcome, and selection of the reported results; Figure 2). Included case series/reports showed a low–moderate quality of evidence, with a total score ranging from 2 to 4 points (Figure 2). Concerning PK studies, quality items were complied for 68.2–95.5% of the total (Figure 3).

## 3. Discussion

To the best of our knowledge, this is the first systematic review that has summarized evidence concerning the PK/PD properties of dalbavancin and its use for in-label or off-label indications in pediatric patients. Overall, the available findings suggested that dalbavancin represents a valuable alternative to daily in-hospital intravenous antibiotic regimens for the management of ABSSSIs or for some off-label indications (i.e., BJIs or CLABSIs) in pediatric scenarios, showing a good efficacy and an excellent safety profile.

In regard to PK properties, dalbavancin exhibited a comparable volume of distribution (15.8 L vs. 15.66 L) and half-life (202 h vs. 174 h) between pediatric and adult patients [1,6,7,21,22,23]. It is noteworthy that pediatric patients showed a slightly lower dalbavancin exposure compared to those observed in randomized comparative studies, including adults affected by ABSSSIs (approximatively 30%) [1,6,7], which is possibly justified by the enhanced renal function that is commonly retrieved in children [24,25]. This results in the adoption of an increased dalbavancin dosing regimen in younger subjects, particularly in those ranging in age from 3 months to 6 years, in order to maximize the attainment of the optimal PK/PD target in terms of the *f*AUC/MIC ratio [19].

In regard to the clinical efficacy and safety profile of dalbavancin for the management of in-label indications (i.e., ABSSSIs) in pediatric patients, the available evidence reported a good efficacy of this long-acting agent, with a clinical success rate above 90%, and no significant difference with traditional anti-Gram-positive agents in a phase III comparative study [8]. Furthermore, the dalbavancin safety profile was excellent in pediatric scenarios, considering that the AE rate was approximatively 10%. Furthermore, it is noteworthy that no case of treatment-related AEs, treatment-related serious AEs, treatment discontinuation due to AEs, or death associated with dalbavancin use were reported in pivotal trials in pediatric patients. Notably, these findings were consistent with those reported in phase III pivotal trials conducted in adult patients receiving a single dose or two dalbavancin doses for the management of ABSSIs, where the clinical success rate exceeded 80% and AEs were reported in approximately 30% of patients, of which only 2% lead to treatment withdrawal [26,27,28].

Concerning dalbavancin use for off-label indications, the available evidence may suggest that this agent could be an effective alternative to daily in-hospital intravenous antibiotic regimens for the management of BJIs or CLABSIs in pediatric patients, based on the overall clinical success rate above 90% coupled with the absence of relapse. An excellent safety profile was also reported for dalbavancin in the setting of real-world studies, considering the low prevalence of AEs, as well as the fact that AEs leading to treatment withdrawal were limited to few cases (namely two cases each of rash/urticaria and vomiting). These findings are consistent with those retrieved in several observational studies conducted among more than 800 adults requiring dalbavancin for different off-label therapeutic indications, in which the overall clinical success rate was above 80% and was associated with limited cases of relapse and/or clinical failure [4,10,11,12,29,30,31,32,33,34,35,36]. Notably, the peculiar PK/PD properties may allow for the management of complicated Gram-positive infections in an outpatient setting, avoiding unnecessary hospitalization and possibly related complications in pediatric patients, similarly to those observed in adults [5].

Given the favorable profile and interest in clinical use, some issues still require attention and possible answers from future studies. Specifically, evidence on the use of dalbavancin for the management of endocarditis or prosthetic vascular graft infections is currently lacking. Furthermore, no studies assessed the tissue penetration rate of dalbavancin in pediatric patients. Notably, according to the difference that emerged in PK studies in terms of serum dalbavancin exposure between pediatrics and adults [6,7], it could not be ruled out that skin/soft tissue, bone, and/or lung dalbavancin absolute concentrations may be affected in pediatric patients [1]. Finally, no cases adopted a therapeutic drug monitoring (TDM)-guided strategy for personalizing dalbavancin dosing regimens in pediatric patients. This may represent an important topic, considering that some evidence supported the clinical usefulness of this approach for harmonizing dalbavancin schedule regimens for subacute and chronic staphylococcal infections, as well as for suggesting dalbavancin redosing for long-term indications [22,37,38,39,40,41,42,43].

Overall, the available evidence supports the valuable role of dalbavancin for the management of both in-label (i.e., ABSSSIs) and off-label indications (i.e., BJIs and CLABSIs) in more than 200 pediatric patients characterized by a wide variability in terms of demographics (age ranging from 3 months to 18 years) and clinical features (the isolation of different Gram-positive pathogens, the adoption of different dalbavancin regimens, and the use of mono vs. combination therapy). Indeed, both the efficacy and safety of dalbavancin treatment was consistent among different demographics and/or clinical subgroups, showing a good clinical cure/response rate (above 90%) coupled with the limited occurrence of AEs leading to treatment discontinuation (less than 10% of the overall included cases). These findings were not only consistent with those retrieved among adult patients treated with dalbavancin for in- and off-label indications, but also with real-life experience reported by a dedicated panel of Italian experts (DalbaPediatriX). Indeed, the panel reported a detailed experience including approximatively fifty pediatric patients treated with dalbavancin in more than ten Italian centers [14,16,17], showing a large variability in terms of age (from neonates to adolescents), underlying disease (including patients with oncological/hemato-oncological diseases, or neuropsychiatric disorders), and indications (including not only in-label indications such as ABSSIs or SSTIs, but also off-label uses such as cases of complicated BJIs, BSIs, endocarditis, or pneumonia). In all of these scenarios, dalbavancin showed its promising role in terms of efficacy and safety profile, which is consistent with that previously reported in adult and pediatric patients, allowing, on the one hand, for the treatment of pediatric patients admitted to emergency wards because of moderate/severe cellulitis and/or skin and soft tissue infections, and, on the other hand, supporting its use in pediatric patients affected by chronic infections (i.e., BJIs) and requiring a long-term and/or suppressive therapy. It is noteworthy that the long half-life of dalbavancin and its good efficacy in staphylococcal infections may provide different peculiar key advantages in childhood when compared to adult patients treated with dalbavancin. Specifically, the use of dalbavancin for in- and off-label indications in childhood may significantly reduce concerns associated with hospital stays, encompassing healthcare cost savings, the decreased risk of nosocomial infections, the mitigation of the discomfort associated with prolonged hospitalization for the children, and the reduced impact on families in terms of quality of life and workday loss. Furthermore, the need of obtaining stable venous access for parenteral daily antibiotic therapy is a challenging concern particularly in neonates, preschool-aged children, or those affected by neuropsychiatric disorders, in which patient sedation is usually required for venous catheter placement, thus also minimizing the risk of the occurrence of medium/long-term complications such as dislocation or catheter-related infections. Some relevant pharmacological issues are associated with the lack of pediatric indications for certain drugs, the limited availability of oral formulations suitable for neonates, the poor compliance with oral antibiotic therapies (e.g., in patients with severe autism who completely refuse to take oral drugs), and the safety concerns associated with prolonged oral therapy particularly with certain agents (i.e., linezolid and/or rifampicin). The benefits provided by dalbavancin in pediatric scenarios may finally result in improved clinical outcomes, also allowing for a significant cost saving (Figure 4). Notably, in the scenario of long-term and/or suppressive dalbavancin treatment, the implementation of a TDM-guided strategy may allow for the personalization of dosing regimens, minimizing the risk of underexposure and failure in attaining optimal PK/PD targets, as previously suggested in adults [41].

The limitations of our systematic review must be acknowledged. Firstly, most of the included studies consisted of case series/reports with a limited sample size; thus, the rate of clinical success should be cautiously interpreted. Additionally, the comparison between dalbavancin and other anti-Gram-positive agents in terms of efficacy and/or safety profile in real-world studies conducted among pediatric patients is lacking. Large heterogeneity among the included studies, as concerns outcome definition, should be mentioned. Furthermore, the number of pediatric patients receiving dalbavancin for off-label indications was limited to less than 30 cases. The search strategy could have failed in entirely retrieving all significant studies concerning the topic, although it was independently performed on at least two large databases, as recommended by PRISMA guidelines. Finally, publication bias cannot be ruled out, and meta-analysis cannot be performed according to the available data. 

## 4. Materials and Methods

We performed a systematic review assessing dalbavancin PK/PD properties and the clinical use for in-label and off-label indications in pediatric patients. The systematic review was conducted according to the Preferred Reporting Items for Systematic Review and Meta-Analyses (PRISMA) guidelines [44] and was registered in the PROSPERO database (CRD42024579252).

### 4.1. Search Strategy

The PubMed-MEDLINE and Scopus databases were independently searched by two investigators (MG and MA) from inception to 20 November 2024, by adopting a specific search string: (“dalbavancin” OR “long-acting antibiotic” OR “long-acting antimicrobial”) AND (“children” OR “pediatric” OR “pediatrics” OR “child” OR “young”). No language limitation was adopted. Furthermore, clinicaltrials.gov was also searched by using the term “dalbavancin” in the field “intervention/treatment”. The retrieved records were checked for duplicates by the same two authors independently. Reference lists of the included studies were screened to identify possible relevant articles. 

### 4.2. Study Selection

The selected studies included RCTs, observational studies, PK studies, case series, and/or reports assessing dalbavancin PK/PD properties or the clinical use for both in-label and off-label indications in pediatric patients. 

Studies were excluded if no quantitative outcome data were, available adult patients were included, and no subgroup analysis for pediatric cohort was provided, or if only in vitro dalbavancin susceptibility was tested from isolates retrieved in pediatric patients and no PK/PD or outcome data were provided. 

Clinical cure/success, as defined in each included study, was selected as the primary outcome. The secondary outcomes were relapse/recurrence rate, microbiological eradication, and the occurrence of adverse drug reactions (ADRs)/adverse events (AEs).

A screening of the titles and abstracts of the retrieved records was independently performed by two authors (MG and MA), and discrepancies were resolved by means of discussion between the two authors or consultation with a third reviewer (DC).

### 4.3. Data Extraction

The relevant data of each included study were independently extracted by two authors (MG and MA) in a pre-specified form. Specifically, the following information was collected for each included study: (a) study author and year of publication; (b) study characteristics (study design, country, time period, sample size, exclusion criteria, and funding sources); (c) demographics and clinical features of included patients (age, sex, site of infection, microbiological isolates, dalbavancin dosing regimen, use of previous antibiotic regimen; adoption of dalbavancin therapeutic drug monitoring [TDM]); and (d) outcome data (clinical cure/success, relapse/recurrence rate, microbiological eradication, occurrence of ADRs/AEs). 

Corresponding authors were contacted in the case of unclear and/or missing data retrieved in the included studies. 

### 4.4. Risk of Bias Assessment

The risk of bias for each included study was independently evaluated by two investigators (MG and MA) for the primary outcome. The Cochrane Risk of Bias Tool (RoB 2.0) [45], the Risk Of Bias In Non-randomized Studies of Interventions (ROBINS-I) [46], and the tool proposed by Murad et al. [47] were adopted for RCTs, observational studies, and case series/reports, respectively. Considering that population PK studies are not interventional in nature and showed an observational single-cohort design, a quality assessment was performed according to the 24-item ClinPK checklist from the Reporting Guidelines for Clinical Pharmacokinetic Studies proposed by Kanji et al. [20]. Potential disagreements were discussed with a third reviewer (DC).

### 4.5. Data Synthesis

Descriptive statistics were used for assessing the primary and secondary outcomes in pediatric patients receiving dalbavancin. The median and interquartile range (IQR) were used for describing continuous variables, whereas categorical variables were reported as absolute values and percentages. A subgroup analysis was performed for describing selected variables according to dalbavancin PK/PD properties and use for in-label or off-label indications.

## 5. Conclusions

In conclusion, our systematic review summarized the evidence concerning the PK/PD properties of dalbavancin and its use for in-label or off-label indications in pediatric patients, suggesting that this agent may be a valuable alternative for the management of ABSSSIs and/or off-label indications in pediatric patients, allowing for a potentially minimization of the duration of hospital stay or even avoiding unnecessary hospitalization. Further prospective studies addressing the efficacy and safety of dalbavancin for in-label and off-label indications in large cohorts of pediatric patients are warranted to support its future widespread use in this challenging scenario.

## Figures and Tables

**Figure 1 antibiotics-14-00121-f001:**
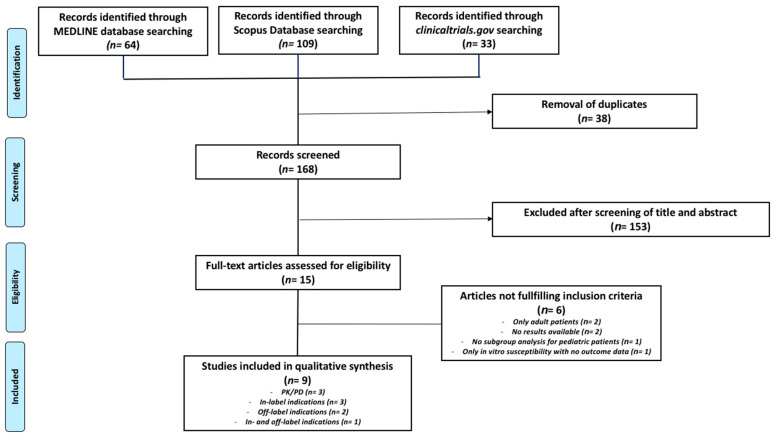
PRISMA flow diagram for study selection.

**Figure 2 antibiotics-14-00121-f002:**
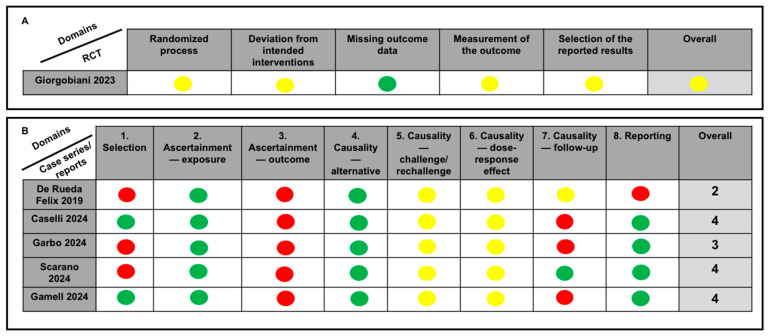
Risk of bias assessment for RCTs (**A**), and case series/reports (**B**) according to specific tools. Green dot: low risk of bias; yellow dot: some concerns; red dot: high risk of bias; gray dot: bias not assessable due to specific features of the study [8,14,15,16,17,18].

**Figure 3 antibiotics-14-00121-f003:**
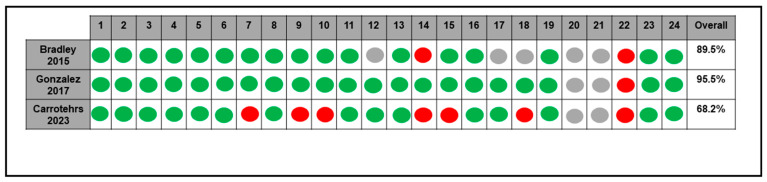
Risk of bias assessment for PK studies according to the tool proposed by Kanji et al. [20]. Green dot: low risk of bias; red dot: high risk of bias [6,7,19].

**Figure 4 antibiotics-14-00121-f004:**
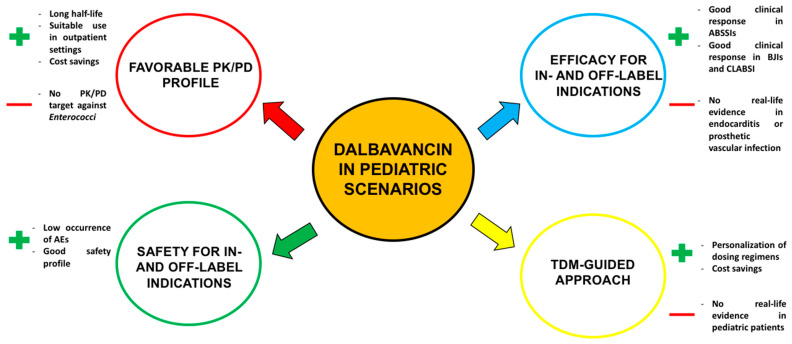
Potential advantages associated with dalbavancin use in pediatric scenarios. ABSSIs: acute skin and skin structure infections; AE: adverse event; BJI: bone and joint infection; CLABSI: central-line-associated bloodstream infection; PK/PD: pharmacokinetic/pharmacodynamic; TDM: therapeutic drug monitoring.

**Table 1 antibiotics-14-00121-t001:** Main features of studies evaluating dalbavancin PK/PD features in pediatric patients.

Study Reference	Study Design	Country	N	Age (Mean or Median)	Male Gender	Dosing Regimens	PK Parameters	PK/PD Relationship	Comparison with Other Populations and Comments
Bradley et al., 2015 [6]	Phase I PK study Multicentric	USA	10	12–17 years	7 M 3 F	1000 mg single dose (>60 kg) or 15 mg/kg single dose (<60 kg)	15 mg/kg group: C_max_: 197.2 mg/L (CV% 26.6) AUC: 16,520 mg*h/L (CV% 20.2) t_1/2_: 202.1 h (CV% 20.0) CL: 11.39 mL/h (CV% 48.2) V_ss_: 11.87 L (CV% 10.7) 1000 mg group: C_max_: 213 mg/L (CV% 11.9) AUC: 18,060 mg*h/L (CV% 28.2) t_1/2_: 227.1 h (CV% 7.3) CL: 16.62 mL/h (CV% 37.4) V_ss_: 15.8 L (CV% 25.5)	N/A	AUC 30% lower compared to adults, according to enhanced renal/hepatic function No significant difference in V_ss_ PTA not assessed
Gonzalez et al., 2017 [7]	Phase I PK study and PopPK model Multicentric	USA	33	3 months-11 years	24 M 9 F	15 mg/kg single dose (age ≥ 5 years) 25 mg/kg single dose (age 2–5 years) 10 mg/kg single dose (age 3 months-2 years)	3 months-2 years group: C_max_: 141 mg/L (114–192 mg/L) AUC: 7890 mg*h/L (6630–11,000 mg*h/L) t_1/2_: 279 h CL: 12.9 mL/h V_ss_: 2.21 L 2–6 years group: C_max_: 328 mg/L (221–443 mg/L) AUC: 22,100 mg*h/L (8670–28,800 mg*h/L) t_1/2_: 315 h CL: 15.8 mL/h V_ss_: 3.30 L 6-11 years group: C_max_: 247 mg/L (183–289 mg/L) AUC: 18,200 mg*h/L (11,500–24,000 mg*h/L) t_1/2_: 390 h CL: 27.7 mL/h V_ss_: 6.93 L	N/A	Higher dosage suggested in children aged 3 months–6 years for enhanced renal function PK comparison with adults not performed PTA not assessed
Carrothers et al., 2023 [19]	PopPK and PK/PD model Multicentric	USA, Bulgaria, Georgia	211 *	3 months-17 years	N/A	N/A	Three-compartment model Serum albumin and creatinine clearance as covariates in the model	A dosing regimen of 22.5 mg/kg in patients < 6 years and of 18 mg/kg in those of 6–18 years provided a PTA ≥ 94% against MRSA with MIC ≤ 2 mg/L (stasis) or ≤0.5 mg/L (2-log kill)	PK comparison with adults not performed

* 204 out of 211 included patients resulted from [6,7,8] AUC: area under time-to-concentration curve; CL: clearance; C_max_: peak concentration; F: female; M: male; MIC: minimum inhibitory concentration; MRSA: methicillin-resistant *Staphylococcus aureus*; N/A: not assessed; PK/PD: pharmacokinetic/pharmacodynamic; PopPK: population pharmacokinetic; PTA: probability of target attainment; t_1/2_: half-life; V_ss_: volume of distribution at steady state.

**Table 2 antibiotics-14-00121-t002:** Main features of studies evaluating dalbavancin use for in-label indications in pediatric patients.

Study Reference	Study Design	Country	N	Age (Mean or Median)	Male Gender	Diagnosis	Isolated Pathogens	Dosing Regimens	Previous Antibiotic Regimen	Dalbavancin TDM	Clinical Cure	Recurrence/ Relapse Rate	AEs
Giorgobiani et al., 2023 [8]	Phase III RCT multicentric	Bulgaria, Georgia	191 (161 vs. 30)	Median: 8 years (range: 0.04–17 years)	119 M 72 F	ABSSSI	90 MSSA 9 *S. pyogenes* 6 MRSA 4 *E. faecalis* 4 *Streptococcus spp* 78 empirical	83 patients: one dose 78 patients: two doses Dosing regimens: 3 months–6 years: 22.5 mg/kg 6–17 years: 18 mg/kg 30 patients in comparator group receiving vancomycin, oxacillin, or flucloxacillin	None	Not performed	Dalbavancin single dose *: 76/78 (97.4%) Dalbavancin two doses *: 73/74 (98.6%) Comparator group *: 26/29 (89.7%)	None	Dalbavancin single dose: 6/83 (7.2%) Dalbavancin two doses: 8/78 (10.3%) In dalbavancin cohorts, pyrexia and cough were reported in more than one patient. No case of treatment-related AEs, treatment-related serious AEs, or AEs leading to discontinuation were reported. Comparator group: 1/30 (3.3%)
Caselli et al., 2024 [14]	Case series	Italy	19	Range: 0.4–18 years	10 M 9 F	ABSSSI (6 complicated)	5 MRSA 3 MSSA 2 *S. pyogenes* 9 empirical	Range total doses: 1–2 (two doses in 4/19 cases) Dosing regimens from 11 mg/kg to 22.5 mg/kg No combination therapy	17/19 (89.5%) VAN 8 DAP 4 BL 3 TEI 2	Not performed	17/19 (89.5%)	None	2/19 (10.5%%; one case of headache and vomiting and one of fever and rash/urticaria; dalbavancin withdrawal in both cases)
Garbo et al., 2024 [17]	Case series	Italy	5	Range: 3 months–13 years	2 M 3 F	ABSSSI	1 CA-MRSA 1 MSSA 1 *S. pneumoniae* 2 empirical	1 dose in all patients Dosing regimens from 18 mg/kg to 22.5 mg/kg No combination therapy	5/5 (100.0%) DAP 3 DAP + CLI 1 BL 1	Not performed	5/5 (100.0%)	None	None
Scarano et al., 2024 [16]	Case series	Italy	4	Range: 2–9 years	2 M 2 F	ABSSSI	2 CA-MRSA 2 empirical	1 dose in all patients Dosing regimens from 18 mg/kg to 22.5 mg/kg Combination therapy with rifampicin in 2 cases	4/4 (100.0%) CLI 3 LIN + RIF 1	Not performed	4/4 (100.0%)	None	None

* 78/83, 74/78, and 29/30 patients treated with one dalbavancin dose, two dalbavancin doses, or comparator agents, respectively, had available clinical outcomes. ABSSSI: acute bacterial skin and skin structure infection; AE: adverse event; BL: beta-lactam; CA-MRSA: community-acquired methicillin-resistant *Staphylococcus aureus*; CLI: clindamycin; DAP: daptomycin; F: female; LIN: linezolid; M: male; MIC: minimum inhibitory concentration; MRSA: methicillin-resistant Staphylococcus aureus; MSSA: methicillin-susceptible *Staphylococcus aureus*; RCT: randomized controlled trial; RIF: rifampicin; TDM: therapeutic drug monitoring; TEI: teicoplanin; VAN: vancomycin.

**Table 3 antibiotics-14-00121-t003:** Main features of studies evaluating dalbavancin use for off-label indications in pediatric patients.

Study Reference	Study Design	Country	N	Age (Mean or Median)	Gender and Ethnicity	Diagnosis	Isolated Pathogens	Dosing Regimens and Combination Therapy	Previous Antibiotic Regimen	Dalbavancin TDM	Clinical Cure	Recurrence/ Relapse Rate	AEs
De Rueda Felix et al., 2019 [18]	Case report	Spain	1	4 years	Female; Caucasian	CAP	N/A	Two doses I dose: 15 mg/kg II dose: 7.5 mg/kg No combo therapy	N/A	Not performed	100%	None	None
Gamell et al., 2024 [15]	Case series	Spain	15	11.9 years (range: 0.3–18 years)	9 Male 6 Female; not specified	5 BJI 4 CLABSI 3 SSI 3 others *	6 MSSA 4 *S. epidermidis* 2 MRSA 1 CoNS 1 *S. lugdunensis* 1 *S. haemolyticus*	Range total doses: 1-16 per patient Dosing regimens from 7.5 mg/kg to 22.5 mg/kg 3/15 cases combination therapy (2 rifampicin and 1 cotrimoxazole)	15/15 (100.0%)	Not performed	13/15 (86.7%)	None	2/15 (13.3%; one case of rash and one of vomiting; dalbavancin withdrawal in both cases)
Caselli et al., 2024 [14]	Case series	Italy	12	Range: 1–17 years	10 Male 2 Female; not specified	BJI	6 MRSA 2 MSSA 1 *S. parasanguinis* 1 *S. hominis + S. epidermidis* 2 empirical	Range total doses: 1–9 Dosing regimens from 17 mg/kg to 22.5 mg/kg No combo therapy	11/12 (91.7%)	Not performed	12/12 (100%)	None	1/12 (8.3%; extravasation)

All three studies retrieved and included for the off-label use of dalbavancin in pediatric patients were no-profit observational studies performed by three different groups of researchers and promoted by respective independent institutions. * one case each of septic thrombophlebitis, chronic cellulitis, and chronic infection of a patch used for repairing congenital heart disease. AE: adverse event; BJI: bone and joint infection; CAP: community-acquired pneumonias; CLABSI: central-line-associated bloodstream infection; CoNS: coagulase-negative Staphylococcus; MRSA: methicillin-resistant Staphylococcus aureus; MSSA: methicillin-susceptible Staphylococcus aureus; N/A: not assessed; SSI: surgical site infection; TDM: therapeutic drug monitoring.

## Data Availability

All data were retrieved from publicly available papers.

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
