# Peer review of "Bridging the Gap: A Systematic Review with Expert Opinion on the Use of Dalbavancin for In-Label and Off-Label Indications in Pediatric Patients"

_antibiotics, 2025, doi:10.3390/antibiotics14020121_

Round 1

Reviewer 1 Report

Comments and Suggestions for Authors

The authors have given a good overview about the PK/PD of dalbavancin.

These are some of the comments on the manuscript:

On page 193, the authors should provide units for dose 1 to 16. For figure 3, the authors could indicate what does the green and red dot represent.

The authors could also elaborate on the adverse events reported in the studies analyzed. 

Author Response

RESPONSE TO REVIEWERS

Manuscript ID: antibiotics-3407333 “Bridge the gap: a systematic review with expert opinion on the use of dalbavancin for in-label and off-label indications in pediatric patients” by Caselli et al.

Dear Editor,

We would like to thank you for the opportunity to resubmit a revised version of this manuscript. We appreciated the reviewer’s constructive comments. All have been carefully considered and incorporated, where and whenever possible, in the revision. Furthermore, as suggested we carefully reviewed the English in order to improve the readability.

Our point-by-point responses are provided below.

Q= QUERY; A= ANSWER

Reviewer #1:

The authors have given a good overview about the PK/PD of dalbavancin.

We thank the reviewer for appreciating our paper.

These are some of the comments on the manuscript:

Q1. On page 193, the authors should provide units for dose 1 to 16. For figure 3, the authors could indicate what does the green and red dot represent.

A1. We thank the reviewer for this comment, allowing us to better clarify this point. In Table 3, the range of total dalbavancin doses is 1-16 per patient, with single dosing regimens ranging from 7.5 mg/kg to 22.5 mg/kg. We specified this issue in Table 3. We also added color legends for Figure 2 and Figure 3 in the footnotes as suggested.

Q2. The authors could also elaborate on the adverse events reported in the studies analyzed.

A2. We thank the reviewer for this relevant comment. We further elaborated on adverse events reported with dalbavancin in the included studies in the Discussion section (refer to Line 231-234 and Line 241-244).

Reviewer 2 Report

Comments and Suggestions for Authors

The authors should consider the followings:

In table 3 (or in text, for the description of table 3), the authors should highlight whether the three major studies were from the same group of researcher or by respective independent institutions. Also , that only a case study with the combined number of patients (less than quantity of 30) may show the very limited knowledge of the field.

In table 1, a column of remarks may be added, to include any special, yet necessary PK/PD content that may not be covered in the exisitng table.

The authors may further identify and discuss the limitations of the article.

In Table 3, the authors may indicate the ethnicity group of the csse patients.

The relevant literatures of the topic should cited.

The color legends should incorporated or clearly explained in Figure 2 and Figure 3.

The article, in principles, followed the PRISMA guideline.

The novelty of the article may be described in the abstract and/or conclusion.

The risk of bias was appropriately used.

Comments on the Quality of English Language

English language editing is needed.

Author Response

RESPONSE TO REVIEWERS

Manuscript ID: antibiotics-3407333 “Bridge the gap: a systematic review with expert opinion on the use of dalbavancin for in-label and off-label indications in pediatric patients” by Caselli et al.

Dear Editor,

We would like to thank you for the opportunity to resubmit a revised version of this manuscript. We appreciated the reviewer’s constructive comments. All have been carefully considered and incorporated, where and whenever possible, in the revision. Furthermore, as suggested we carefully reviewed the English in order to improve the readability.

Our point-by-point responses are provided below.

Q= QUERY; A= ANSWER

Reviewer #2:

The authors should consider the followings:

Q1. In table 3 (or in text, for the description of table 3), the authors should highlight whether the three major studies were from the same group of researcher or by respective independent institutions. Also , that only a case study with the combined number of patients (less than quantity of 30) may show the very limited knowledge of the field.

A1. We thank the reviewer for this comment, allowing us to better clarify this issue. It should be noticed that all the three studies retrieved for off-label use of dalbavancin in pediatric patients (references No. 14-15 and 18) and reported in Table 3 were no-profit observational studies performed by three different groups of researchers and promoted by respective independent institutions. We better clarify this point in the footnotes of Table 3. Furthermore, we added among limitations of our systematic review the fact that the number of pediatric patients receiving dalbavancin for off-label indications was limited (refer to Discussion section; Line 321-323)

Q2. In table 1, a column of remarks may be added, to include any special, yet necessary PK/PD content that may not be covered in the exisitng table.

A2. We thank the reviewer for this suggestion. We added in the last column of Table 1 specific comments concerning PK/PD features not assessed in included studies.

Q3. The authors may further identify and discuss the limitations of the article.

A3. We thank the reviewer for this comment. We extended limitations section of our paper (refer to Discussion section; Line 316-326).

Q4. In Table 3, the authors may indicate the ethnicity group of the case patients.

A4. We thank the reviewer for this suggestion. We added the ethnicity group of included patients in Table 3. Although all the three studies were conducted in Europe (two in Spain and one in Italy), ethnicity of included patients was specified only in the case report (reference no. 18).

Q5. The relevant literatures of the topic should cited.

A5. We thank the reviewer for this comment. Relevant literatures on the topic have been cited and detailed according to a systematic search strategy independently performed on two different databases (refer to Materials and Methods section, Line 334-342), which allowed to entirely retrieved all evidence concerning the use of dalbavancin in pediatric patients, as summarized in Table 1-3 (references No. 6-8 and 14-19). Furthermore, relevant literatures have been cited for discussing retrieved evidence (references No. 20-42).  

Q6. The color legends should incorporated or clearly explained in Figure 2 and Figure 3.

A6. We thank the reviewer for this suggestion. We added color legends for Figure 2 and Figure 3 in the footnotes.

Q7. The article, in principles, followed the PRISMA guideline.

A7. We thank the reviewer for this comment. As recommended by Cochrane guidelines, the PRISMA guideline has been followed for performing our systematic review. This issue is specified in the Materials and Methods section (refer to Line 328-332). Furthermore, PRISMA checklist is reported among Supplementary Materials.

Q8. The novelty of the article may be described in the abstract and/or conclusion.

A8. We thank the reviewer for this comment, allowing us to better clarify the novelty of our article. We reported the novelty of our article in both Abstract (refer to Line 39-40) and Conclusion sections (refer to Line 391-392).

Q9. The risk of bias was appropriately used.

A9. We thank the reviewer for this comment. As recommended, the risk of bias was assessed by using the recommended tools currently available according to the Cochrane guidelines, namely the RoB 2.0 for RCTs and the ROBINS-I for observational studies. In regard to case series/reports and PK studies the accepted and previously used tools proposed by Murad et al. (reference No. 46) and Kanji et al. (reference No. 47) have been implemented. This issue is specified in the Materials and Methods section (refer to Line 372-381).

Reviewer 3 Report

Comments and Suggestions for Authors

Hi, 

This was a nice read and an addition to my knowledge about DAL. 

Thank you and all the best. 

Bridge the gap: a systematic review with expert opinion on the use of dalbavancin for in-label and off-label indications in pediatric patients

Summary:

The review article focused on use of dalbavancin an important treatment option against Gram positive pathogens. The authors have nicely described the use of DAL in pediatric populations ranging from 3 months to 18 years. The authors based on the review have identified that the use of DAL against AABSSSIs in pediatric population could be an viable option to consider. They also suggested use of DAL in off-label indications. This report could be useful for clinicians and those who further plan to conduct some clinical trials using DAL. There are some minor comments which can be addressed to enhance thee article.

Overall comments

1.       Line 48: If possible, it would be nice to have a MIC range for these pathogens

2.       Line 52: if fAUC/MIC is the good predictor of efficacy can you include the range of AUC/MIC, it would be good to see this in adults and in pediatrics (based on data availability). Also kindly include your suggestions or interpretations based on the available data for an targeted AUC/MIC ratio for pediatric population considering changes in the PK between adults and children.

3.       Line 57-62: What is the status of this drug in the US, is it used or not.

4.       Line 85: Define RCT, I see it defined in the abstract and in the method section which appears at the end of the manuscript.

5.       Table 1: Recommend looking in the article PMID: 39299696. There is only one study where the PTA was assessed as a PD marker. Aren’t there any other studies which mention PD readout.

6.       Line 114-119: This is also a POP PK model and the data about dose 18 -22.5 mg /kg is based on model simulations, please confirm and change accordingly. Suggest rearranging the dosing based on the youngest to oldest and comment if there was significant difference in the weights.

7.       Table 2: Suggest including the prior antibiotic treatments which could be useful for physicians.

8.       Table 2: Kindly include the details of AEs for first study as indicated for others.

9.       Line 143-144: Replace “Dosing” with “Dose”

10.   Line 203 : This only describes the PK so remove the PD from PK/PD.

Overall this was a good read and will be hepful to open new avenues for use of Dalbavancin as a treatment for acute bacterial skin and skin structure infection (ABSSSIs) in pediatric population.

Decision: Accepted with corrections.

Author Response

RESPONSE TO REVIEWERS

Manuscript ID: antibiotics-3407333 “Bridge the gap: a systematic review with expert opinion on the use of dalbavancin for in-label and off-label indications in pediatric patients” by Caselli et al.

Dear Editor,

We would like to thank you for the opportunity to resubmit a revised version of this manuscript. We appreciated the reviewer’s constructive comments. All have been carefully considered and incorporated, where and whenever possible, in the revision. Furthermore, as suggested we carefully reviewed the English in order to improve the readability.

Our point-by-point responses are provided below.

Q= QUERY; A= ANSWER

Reviewer #3:

Hi, This was a nice read and an addition to my knowledge about DAL. Thank you and all the best.

Bridge the gap: a systematic review with expert opinion on the use of dalbavancin for in-label and off-label indications in pediatric patients

Summary:

The review article focused on use of dalbavancin an important treatment option against Gram positive pathogens. The authors have nicely described the use of DAL in pediatric populations ranging from 3 months to 18 years. The authors based on the review have identified that the use of DAL against AABSSSIs in pediatric population could be an viable option to consider. They also suggested use of DAL in off-label indications. This report could be useful for clinicians and those who further plan to conduct some clinical trials using DAL. There are some minor comments which can be addressed to enhance the article.

We thank the reviewer for appreciating our manuscript.

Overall comments

Q1.       Line 48: If possible, it would be nice to have a MIC range for these pathogens

A1. We thank the reviewer for this comment. We added the MIC50 and MIC90 values for MRSA as suggested (refer to Line 51-52).

Q2.       Line 52: if fAUC/MIC is the good predictor of efficacy can you include the range of AUC/MIC, it would be good to see this in adults and in pediatrics (based on data availability). Also kindly include your suggestions or interpretations based on the available data for an targeted AUC/MIC ratio for pediatric population considering changes in the PK between adults and children.

A2. We thank the reviewer for this comment, allowing us to better clarify this issue. As suggested, we added the fAUC24/MIC ratio required for ensuring a 2-log-kill activity (i.e., fAUC24/MIC >111.1) against Staphylococcus aureus as reported in preclinical models (refer to Line 56). Notably, this best PK/PD target is the same for adult and pediatric population. On the other hand, considering the different PK behavior of dalbavancin in pediatrics compared to adults, specifically a reduction of 30% in AUC as reported in PK studies, an increased dalbavancin dosing regimen has been scheduled in pediatric patients in order to maximize the attainment of the best PK/PD target in terms of fAUC/MIC ratio. We better clarify this issue in the Discussion section (refer to Line 219-225).

Q3.       Line 57-62: What is the status of this drug in the US, is it used or not.

A3. We thank the reviewer for this comment, allowing us to better clarify this important topic. Dalbavancin was approved by the Food and Drug Administration for the management of ABSSSI both in adults and in pediatrics in 2014 and 2021, respectively. We added this information in the Introduction section (refer to Line 61-66).

Q4.       Line 85: Define RCT, I see it defined in the abstract and in the method section which appears at the end of the manuscript.

A4. Thank you for this suggestion. We defined RCT at first mention in the Results section (refer to Line 90).

Q5.       Table 1: Recommend looking in the article PMID: 39299696. There is only one study where the PTA was assessed as a PD marker. Aren’t there any other studies which mention PD readout.

A5. We thank the reviewer for this comment. Unfortunately, the abovementioned study (One and Done? An Evaluation of the Clinical Outcomes of Single- and Multi-Dose Dalbavancin Use in the Pediatric Population; doi: 10.1177/08971900241285521) was not retrieved with our search strategy, and cannot be included in our systematic review. We added this issue among limitations of our study (refer to Line 323-325). In regard to Table 1, to the best of our knowledge and according to our search strategy only the population PK and PK/PD model reported in the Reference No. 19 evaluated PTA against different staphylococcal MIC values, and no other studies were retrieved.

Q6.       Line 114-119: This is also a POP PK model and the data about dose 18 -22.5 mg /kg is based on model simulations, please confirm and change accordingly. Suggest rearranging the dosing based on the youngest to oldest and comment if there was significant difference in the weights.

A6. We thank the reviewer for this comment, allowing us to better clarify this issue. As suggested, we modified the sentence accordingly, by specifying that the study reported in Reference No. 7 is a population PK study and that the data about dosing regimens of 18-22.5 mg/kg were based on model simulations (refer to Line 119-127). Furthermore, we rearranging the dosing based on the youngest to oldest, including also data on median weight of each cohort as suggested (refer to Line 119-127).

Q7.       Table 2: Suggest including the prior antibiotic treatments which could be useful for physicians.

A7. We thank the reviewer for this relevant suggestion. We added prior antibiotic treatments in studies reported in Table 2.

Q8.       Table 2: Kindly include the details of AEs for first study as indicated for others.

A8. We thank the reviewer for this relevant suggestion. We detailed AEs in dalbavancin cohorts also for study reference no. 8.

Q9.       Line 143-144: Replace “Dosing” with “Dose”

A9. Thank you for this suggestion: we replaced the term “dosing” with “dose” (refer to Line 152-153).

Q10.   Line 203 : This only describes the PK so remove the PD from PK/PD.

A10. Thank you for this suggestion: we removed the PD considering that only PK features have been described (refer to Line 217).

Overall this was a good read and will be helpful to open new avenues for use of Dalbavancin as a treatment for acute bacterial skin and skin structure infection (ABSSSIs) in pediatric population.

Decision: Accepted with corrections.

Thank you for the appreciation.